# Study on the High-Temperature Interaction between Coke and Iron Ores with Different Layer Thicknesses

**DOI:** 10.3390/ma17061358

**Published:** 2024-03-15

**Authors:** Yong-Hong Wang, Ping Du, Jiang Diao, Bing Xie, Ming-Hua Zhu

**Affiliations:** 1Institute Research of Iron & Steel of Sha-Steel, Zhangjiagang 215625, China; duping@shasteel.cn (P.D.); zhumh-iris@shasteel.cn (M.-H.Z.); 2School of Materials Science and Engineering, Chongqing University, Chongqing 400044, China

**Keywords:** coke layer thickness, cohesive zone, permeability, blast furnace

## Abstract

Coke plays a key role as the skeleton of the charge column in BF. The gas path formed by the coke layer in the BF has a decisive influence on gas permeability. At high temperatures, the interface between coke and ore undergoes a melting reaction of coke and a reduction reaction of ore. The better the reducibility of the ore, the more conducive it is to the coupling reaction of ore and coke. The melting loss reaction of coke becomes more intense, and the corresponding strength of coke will decrease, which will affect the permeability of the blast furnace and is not conducive to the smooth operation of the blast furnace. Especially with a deterioration in iron ore quality, BF operation faces severe challenges, which makes it necessary to find an effective way to strengthen BF operation. In this study, a melting-dropping furnace was used to develop and clarify the high-temperature interaction between coke and iron ores with different layer thicknesses. The influencing factors were studied by establishing a gas permeability mathematical model and observing the metallographic microscope images of samples after the coke solution loss reaction. The relationships between coke layer thickness, distribution of gas flow, and pressure drop were obtained. The results showed that, under certain conditions, the gas permeability property of a furnace burden has been improved after the coke layer thickness increased. Upon observing the size of coke particles at the interface reaction site, the degree of melting loss reaction can be determined. A smaller particle size indicates more melting loss reaction. A dripping eigenvalue for molten metal was introduced to evaluate the dynamic changes in the comprehensive dripping properties of molten metal of furnace burden, which showed that the dripping eigenvalue for the molten metal could deteriorate because of the unruly thickness and the coke layer thickness should be limited through considering the operational indicators of the blast furnace.

## 1. Introduction

Blast furnace technology is still the main ironmaking route, with a current global share of 70% [1]. Coke is the main material in blast furnaces, and it plays an important role in blast furnace operation. Coke functions not only as a reducing agent but also as a skeleton in the blast furnace [2]. The blast furnace is filled with furnace burden, molten slag, iron, and coke, which account for 1/3–1/2 of the furnace burden and determines the gas permeability [3,4].

The dissected blast furnace [5] showed that the coke solution loss reaction influenced the gas permeability of the upper part of BF [6,7,8], and there are many references available in this field [9,10]. In addition, the thickness of the coke layer is also an important factor affecting the distribution of gas flow. Unsuitable coke layer thickness may cause an unstable gas flow distribution as well as a central gas flow block, which threatens BF operation. 

The coke layer thickness is a restrictive factor for normal BF production and raising the ratio of coal. It will affect the size of the coke window, the amount of CO produced, the permeability of the material layer, and the utilization rates of gas and heat energy. As the thickness of the coke layer increases, the degree of ore–coke mixing decreases, and the resistance generated by interface reactions decreases. At the same time, as the area of a single coke window increases, the stability and uniformity of gas permeability are improved, and the pressure is reduced. So, it is necessary to find the suitable coke layer thickness to optimize the BF index and its significant meanings and practical application value to ensure the stability of actual production. 

In the past, most of the research on coke mainly focused on the type of coke, the ratio of mixed coke and the size of coke in the ore layer, and the influence of coke thickness on permeability is also discussed. However, due to the limitation of equipment, there are few literature studies about the influence of coke layer thickness on the drop performance of blast furnace burden. In fact, there were complicated interactions among the furnace burden, which could improve molten-metal dripping properties distinctly if effectively used [11,12]. Increasing the coke layer thickness could improve the permeability of BF, which resulted in an improvement in the dripping properties of molten metal. The Shougang Jingtang 2# blast furnace achieved long-term stable smelting of a large ore batch with a large coke ratio (thick coke layer) by using AB ore in the form of a small material volume [13]. However, the coke layer thickness cannot be increased indefinitely. Too thick a coke layer could generate huge resistance to the process of molten iron dripping, which is one of the important causes that lead to irregular furnace operation. The studies [14,15] show that an increase in the thickness of the coke layer means an increase in the thickness of the material layer, an increase in the “gas window” area of each layer of the soft melt zone, and an improvement in the permeability of the blast furnace. However, as the thickness of the blast furnace ore and coke layer increases, the pressure difference in the soft melt zone increases, and the total pressure difference slightly decreases to a certain point and then increases again. 

In this paper, research on the interaction between coke and iron ores was based on the melting-drop furnace. The experiment of solution loss reaction of coke was carried out, and a mathematical model of gas permeability was established. To clarify the mechanism of the interaction, the dynamic changes in the influences of coke layers with different thicknesses on the permeability were investigated through the combination of experimental results and mathematical models, and then the influence of the thickness on the melt-dropping property of the burden was studied. 

## 2. Materials and Experiment

### 2.1. Raw Iron Ores and Coke

The raw materials were iron ores and coke for field production, including sinter (S-1), pellets (P-1), and 2 lumps (O-1, O-2). The chemical compositions are listed in Table 1, and the quality of the coke is listed in Table 2. Quantitative analysis of the chemical composition of iron ore analyzed in accordance with GB/T 6730.87-2023 [16] Iron ores—determination of total iron and other multi-element content—wavelength dispersive X-ray fluorescence spectrometry (cobalt internal standard method), and the analysis of FeO is analyzed in accordance with GB/T 6730.8-2016 [17] determination of ferrous content in iron ores—potassium dichromate titration method. The composition analysis of coke shall be determined in accordance with GB/T 2001-2013 [18] coke industrial analysis method, M40 shall be determined in accordance with GB/T 2006-2008 [19] coke mechanical strength determination method, coke CSR shall be determined in accordance with GB/T 4000-2017 [20] coke reactivity and post reaction strength test method, and coke Ms shall be determined in accordance with ISO 728-2021 [21] size analysis by sieving. Because different chemical compositions of ores can lead to differences in the internal structure of minerals, affecting their reducibility, for eliminating the influence of different ore structures on the interfacial reaction between ore and coke, O-1 and O-2 with similar compositions but different origins, and different structures were selected to test the contrast.

In Table 2, M40 is the crushing strength of coke. CSR is coke strength after the reaction. Ms is the average particle size of coke. A is coke ash content. S is coke sulfur content. C is the fixed carbon content of coke.

### 2.2. Development of Melting-Drop Furnace

The working principle in a melting-drop furnace is shown in Figure 1. There are two parts of the equipment, which are the gas control part and the reacting part. The gas used in the experiment was mixed by the gas flow controller and injected from the side of the furnace body. After contacting the material layer, the gas passes through the material layer vertically. The pressure was measured by a pressure gauge, the temperature measured by a thermocouple, and the shrinkage rate of the sample bed was gathered by a control box for calculation and drawing curves. The sample bed, which consists of alternately packed ore and coke layers, was arranged in the reacting parts, as shown in Figure 2.

So, for each test, iron ore and coke with a layer thickness were placed in a graphite crucible (Figure 2), the inner diameter of which was 120 mm; there was a graphite gasket of the same size at the bottom on which samples could be placed; and 5-mm holes were evenly distributed on the graphite gasket, from which the drops will drop at high temperature. The maximum load of 0.35 MPa was applied to the sample bed from the top by the rod. The generated molten iron was collected in the sample mouth. The devices mentioned above were cleverly designed so that the gas could only pass through the packed bed without leakage. Then, the research on the influence of coke layers with different thicknesses on the high-temperature interaction between coke and iron ore could be carried out on this set of equipment.

In the experiment, the particle size range of the sample, the temperature, and the gas composition shall be executed in accordance with the national standard GB/T 34211-2017 [22]. The diameters of coke and ore in this experiment are 10–12.5 mm. Because the particle size ranges of ore and coke are unified, it can effectively avoid the impact of original particle size segregation on experimental results. In addition, the amount of coke and ore samples is determined based on the size of the crucible in the equipment.

## 3. High-Temperature Interaction between Coke and Iron Ores

### 3.1. Experimental Conditions

The experimental conditions of material charging are shown in Table 3.

In each experimental condition, the total amount of iron ore and coke is kept at 500 g and 80 g, respectively. The thickness of iron ore and coke is 160 mm, and the thickness of the coke layer was changed to 20 mm, 30 mm, and 40 mm. Moreover, 80 g coke can guarantee the amount when the thickness of the coke layer changes from 20 mm to 40 mm.

Figure 3 shows the experimental conditions for the temperature and gas composition. In order to simulate the melting behavior of iron ore in BF, the burden was heated to 900 °C in an N_2_ atmosphere, then the heating rate was slowed, and the gas composition was changed to simulate the environments of both.

### 3.2. Experimental Results

#### 3.2.1. Dripping Properties of Molten Metal

In this study, it was impossible to observe the packed bed behavior during the high-temperature part; therefore, to evaluate the interaction between the coke and iron ores, the measured parameters were quantified. T_10_ and T_40_ were the temperatures when the packed bed shrinkage was 10% and 40%, respectively. T_s_ was the temperature when the pressure increased steeply, and P_max_ was the highest value of the pressure. The S value was the integral of the pressure drop value over the cohesive zone, which reflected the permeability of the cohesive zone. The equation of S value is shown as Equation (1).
(1)$$S=\int_{T_{s}}^{T_{p}}(P_{t}-Ps)dt$$

The testing results of dripping properties are listed in Table 4, and in order to maintain the reproducibility of the experimental results, each group of experiments was repeated twice. As the results show, the properties of furnace burden have been distinctly changed due to the change in coke layer thickness, embodied in softening temperature, softening interval, melting temperature, pressure drop, and S value.

#### 3.2.2. Pressure Drop in Layer Thickness Condition

Figure 4 reflects the relationship between the pressure drop and the temperature in different conditions of coke layer thickness. The value of Ts increased in different degrees when the coke layer thickness increased, and that of five kinds of furnace burden increased to 1295.9 °C, 1289 °C, and 1300 °C, respectively. Increasing coke layer thickness contributed to raising the melting temperature and meeting the demand for a higher melting temperature during BF operation.

In the soft melt droplet layer, as the thickness of the coke layer increases, the skeleton effect of the coke strengthens, the permeability of the soft melt droplet layer improves, and the maximum pressure drop decreases. At the same time, as the thickness of the coke layer increases, with an increase in the ore layer thickness, the homogeneity of the ore coke weakens relative to the soft melting zone, the interaction between the ore coke weakens, and the softening and dripping temperatures both increase [23].

In addition, the coke layer is too thin, and due to the viscosity of the molten metal, it is easy to cause a decrease in permeability after infiltration into the coke layer, leading to fluctuations in gas flow. So, increasing the thickness of the coke layer can improve air permeability.

In Case 2 and Case 3, the pressure drop decreased distinctly due to the change in the coke layer. P_max_ decreased from 9322 Pa to 6558.7 Pa and 4970 Pa. Compared with Case 4 and Case 5, the pressure peak value decreased by 2206 Pa.

#### 3.2.3. Shrinkage Rate in Layer Thickness Condition

The relationship between shrinkage rate and temperature in different conditions of layer thickness is shown in Figure 5. Under the same charging condition, with an increase in the thickness of the coke layer from 20 mm to 30 mm and 40 mm, T_10_ decreased obviously, while T_40_ did not change greatly. The initial softening temperatures decreased by about 22 °C and 28 °C, respectively.

#### 3.2.4. Effect of Coke Solution Loss

Figure 6 presents the dissected graphite crucible in conditions of different thicknesses, and the metallography images of the observed section are shown in Figure 7.

As shown in Figure 7, at the cross-sectional area of the surface of the coke layer and pig iron, there were many coke particles embedded in the pig iron. These particle sizes were inhomogeneous, ranging from 30 μm to 300 μm. The amount of coke particles was larger when the coke layer thickness was 20 mm than when it was 30 mm or 40 mm. Moreover, the particle sizes were far smaller at 40 mm than at 20 mm and 30 mm. That may illustrate the mechanical strength of the coke layer, which was damaged seriously due to smaller thickness and the wrecked coke particles mixed with molten iron.

## 4. Discussion

### 4.1. Solution Loss Reaction of Coke

The solution loss reaction of coke was affected by changing the thickness of the coke layer, which could wreck the coke in the cohesive zone and dropping zone. Since the downward movement of the modified coke is not conducive to gas–liquid infiltration of the lower part of the blast furnace, the solution loss reaction might reduce the utilization of chemical and thermal energy of BF gas, which is expressed by Equation (2).
(2)$${CO}_{2}+C(s)=2CO$$

The reduction reaction of iron oxide is presented by Equation (3), where the iron oxide was expressed by FeO_x_ to facilitate analysis.
(3)$${FeO}_{x}+CO={FeO}_{x-1}+{CO}_{2}$$

Through Equations (2) and (3), Equation (4) is obtained, which presents the coupling reaction between the solution loss reaction of coke and the reduction reaction of iron oxide.
(4)$${FeO}_{x}+C={FeO}_{x-1}+CO$$

Equation (4) could be a cycle on the utilization of CO and CO_2_. It can be seen from the equation that increasing the thickness of the coke layer could promote the reaction and generate more CO, which facilitates the reduction reaction.

In addition, the reaction that Equation (4) presents has a great influence on the mechanical strength of the coke layer. The research has shown [24] that at 1100 °C the reaction mode of coke is mainly based on permeation reaction, and with an increase in melting loss rate, the damage to the internal structure of coke increases, leading to a sharp decrease in CSR; for every 1% increase in melt loss rate, CSR decreases by about 1.38%. However, when the temperature exceeds 1350 °C, the melting loss reaction mainly concentrates on the outer surface of the coke, causing less damage to its internal structure. As the melting loss rate increases, the CSR does not change much; for every 1% increase in melt loss rate, CSR decreases by about 0.63%. The skeleton function of the coke layer would be seriously wrecked due to the thinner coke layer thickness, which would result in the blockage of the gas flow path and the destruction of the permeability of BF. The thickening of the coke carbon layer reduces the number of interfaces between the ore and coke, the pressure loss of the gas through the mixed layer, especially the block zone, decreases, the interface effect between the material surface decreases, the ‘coke window’ is more stable, the upper air flow distribution of the blast furnace is more reasonable, and the gas flow is more uniform in the circumferential distribution of the throat, which is conducive to the stable and smooth operation of the blast furnace.

### 4.2. Coke Layer Function in Cohesive Zone

The cohesive zone is the transition part between the lumpish zone and the dropping zone. The weak permeability of the cohesive zone may block the upward movement of gas flow at the center and develop edge gas, resulting in better permeability of the periphery than at the center of the cohesive zone. The pressure drop of BF is the sum of the pressure drop of the lump charging zone, cohesive zone, and dripping zone, as expressed by Equation (5).
(5)$$∆P=∆P_{L}+∆P_{cz}+∆P_{d}$$

As Figure 8 shows, the coke layer formed dendritic coke windows in the BF interior, and the gas flow distribution secondly turned center distribution into edge distribution. The pressure drop in the cohesive zone is calculated by Equation (6).
(6)$$∆P_{cz}=∆P_{o}+∆P_{c}+∆P_{s}$$

The coke windows would be destroyed when the coke layer was too thick, and the immediate increase in ΔPcz and ΔPc would lead to the disorder of gas flow distribution. In addition, the mechanical strength of coke was wrecked due to the solution loss reaction of coke, which further deteriorated the permeability. 

In addition, the desolvation reaction of coke destroys the mechanical strength of coke and further deteriorates the permeability. Increasing the thickness of the coke layer is conducive to protecting the thickness of the ‘coke window,’ narrowing the height of the soft melt zone, improving the permeability of the overall material column, and better playing the role of the coke skeleton.

### 4.3. Mathematical Model of Permeability

The critical factor determining the normal operation of BF is ΔP/H. In order to quantify the permeability of the furnace charge, the permeability of burden was calculated by the Ergun equation [25], which is shown by Equation (7), where d_p_, u, ε, ρ, μ, φ were defined as particle diameter, gas velocity, void fraction, density, viscosity, and the shape factor, respectively. Furthermore, the coke average diameter was 10 mm, the void fraction was 0.418, and the shape factor was 0.785, as is shown in the research of Yamada [26].
(7)$$\frac{∆P}{L}=150\frac{{\left(1-\varepsilon \right)}^{2}}{\varepsilon^{3}}\cdot \frac{\mu \cdot u}{{\left(\varphi \cdot d_{p}\right)}^{2}}+1.75\frac{\left(1-\varepsilon \right)}{\varepsilon^{3}}\cdot \frac{\rho u^{2}}{\left(\varphi \cdot d_{p}\right)}$$

Most of the gas flow in the cohesive zone may occur as lateral flow due to the great resistance in the cohesive zone, which is because of the small porosity resulting from the molten iron. Therefore, the ΔP/H could be analyzed by Equation (8).
(8)$$\frac{△P}{H}=K\rho \omega^{2}\frac{L^{0.183}}{n^{0.46}\cdot h_{c}^{0.93}\cdot \varepsilon^{3.74}}$$
where K, ρ, ω, L, n, h_c_, and ε were defined as a constant coefficient, gas flow density, velocity of gas flow in an uncharged furnace, cohesive interval, amount of coke layers, coke layer thickness, and void fraction of coke, respectively. 

In this paper, as reported by Yamada [19], the density of gas flow is considered as 1.25 kg·m^3^, and the velocity of gas flow is 1.02 m·s^−1^. Furthermore, the void fraction is 0.418. As is shown in Equation (8), all of the influencing factors in the equation are the greatest influence due to power exponents. After the quantification of influencing factors, the resistance on gas flow in the cohesive was calculated. It is shown that the thickness of the coke layer, as well as the amount of the coke layer, are the determining factors of ΔP/H.

To confirm this conclusion, the calculation results of Equation (7) and the measurement results are compared, and the contrast results are shown in Figure 9. It can be seen that the results of the calculation show good agreement with the measurement result, which indicated that a higher thickness of the coke layer is beneficial to improve the permeability of the cohesive zone. That is because an increase in the thickness of the coke layer could lead to the diversion of the gas flow and make coke play a better role as “shutters”.

However, the BF is a synthesis reactor that is affected by temperature, pressure drop, BF burden, etc. It is not always possible to optimize the operation of a blast furnace by increasing the thickness of the coke layer. According to Table 4, the S value of the coke layer in thickness of 30 mm is the best of all the schemes, and it indicates that the coke layer thickness can not increase indefinitely. The relationship between the S value and the thickness of the coke layer was simulated using a mathematical model, and the fitting curve is listed in Figure 10.

It can be seen from Figure 10 that the relationship between the S value and the thickness of the coke layer is a quadratic function. When the layer thickness is higher than 30 mm, the influence of an increase in the coke layer thickness on the S value is completely opposite to that before. This is because the thickness of the coke layer increases too much so that the path inside the coke can be extended. In the dropping process of molten iron, the permeability of the coke layer decreases due to the excessive infiltration of molten iron. As a result, the permeability of the burden began to decline, which has a great resistance to the gas flow, thus hindering the dropping process of molten iron. Related studies have shown [27] that measures to improve the permeability of the soft melt zone can be taken by expanding the ore and coke batches to increase the thickness of the coke layer inside the soft melt zone, improve reducibility, and suppress interaction to reduce the radial width of the soft melt zone while improving the permeability of the soft melt layer. Therefore, expanding the thickness of the coke layer must have a ‘degree.’ When increasing the thickness of the coke layer, effective measures should be taken to open the center to prevent the deterioration of the permeability of the material column.

Shorten the time difference between iron phase melting and slag phase melting so that a large amount of slag phase melts and infiltrates the coke layer during iron phase melting, resulting in the phenomenon of melting and dripping of the furnace material. This state should be the focus of future research on blast furnace material structure, loading system, lower blast system, and oxygen enrichment, with the expectation of obtaining reasonable process parameters to achieve the above state.

## 5. Conclusions

In this study, the melting-drop furnace was developed, and the experimental methods according to national standards were used to explore the high-temperature interaction between coke and iron ores with different layer thicknesses. The mathematical model of permeability and coke solution loss reaction was introduced for analysis. The following conclusions were made when coke ratio is constant:The solution loss reaction of coke and reduction reaction of iron oxide could be a recycling of CO and CO_2_. Raising the thickness of the coke layer could promote this coupling reaction and increase the CO concentration, which is beneficial for indirect reduction. The mechanical strength of the coke layer was seriously damaged when the thickness of the coke layer became thinner. However, further quantitative research is still needed.Coke layers with higher thickness are propitious to the development of the dripping properties of molten iron. In this study, the layer thickness of coke was adjusted from 20 mm to 30 mm and 40 mm, where the permeability of the cohesive was improved to some extent, but the best is 30 mm.The excessively higher coke layer extended the dropping distance of iron as well as the contact time between molten iron and coke, resulting in huge resistance for the dripping process of molten iron. Then, the dripping properties of molten metal could be wrecked. Therefore, it is necessary to choose the appropriate coke layer thickness based on the volume of the blast furnace and the requirements for production.Increasing the thickness of the coke layer is one of the main measures to strengthen smelting, reduce the coke ratio, and stabilize the upper gas flow. A reasonable thickness of the coke layer plays a crucial role in the rational distribution of the furnace charge and upper gas flow, as well as in improving the utilization of gas heat energy, improving the permeability of the blocky zone, and ensuring the smooth descent of the furnace charge, uniform, and active operation of the furnace charge column, and the reasonable distribution of initial gas flow.

## Figures and Tables

**Figure 1 materials-17-01358-f001:**
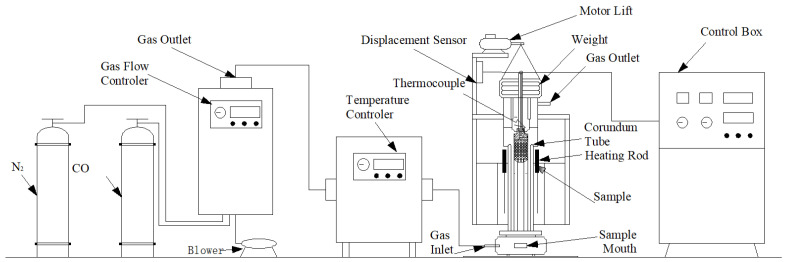
Schematic drawing of large capacity melting-drop furnace.

**Figure 2 materials-17-01358-f002:**
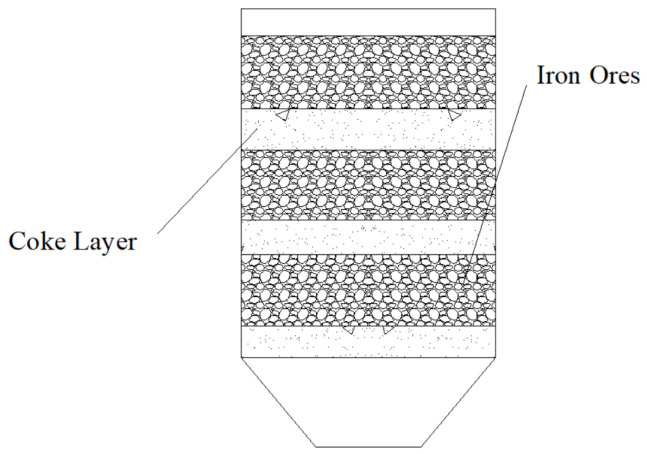
Diagram of raw materials charging.

**Figure 3 materials-17-01358-f003:**
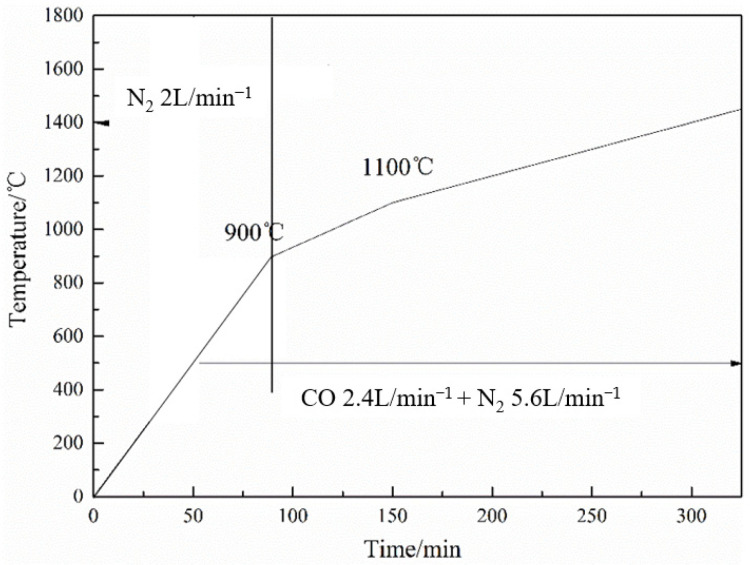
Experimental conditions of the temperature and gas composition.

**Figure 4 materials-17-01358-f004:**
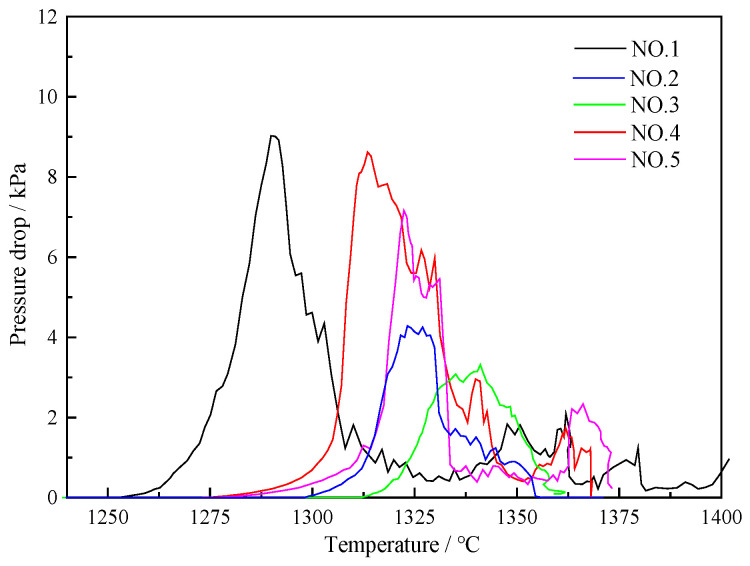
Relationship between pressure drop and temperature.

**Figure 5 materials-17-01358-f005:**
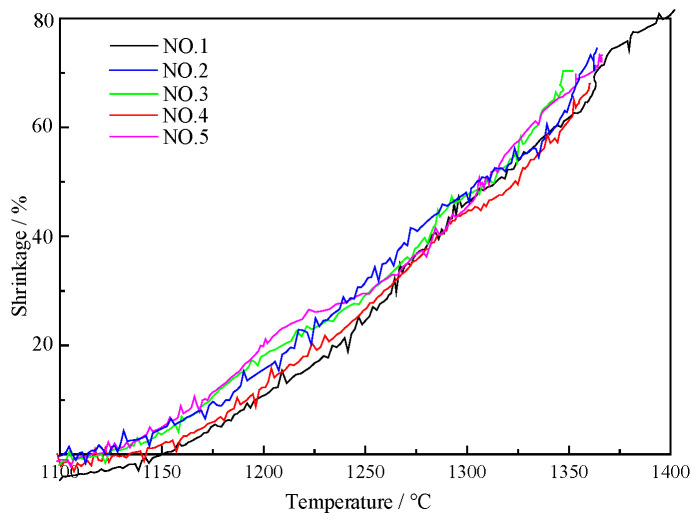
Relationship between shrinkage rate and temperature.

**Figure 6 materials-17-01358-f006:**
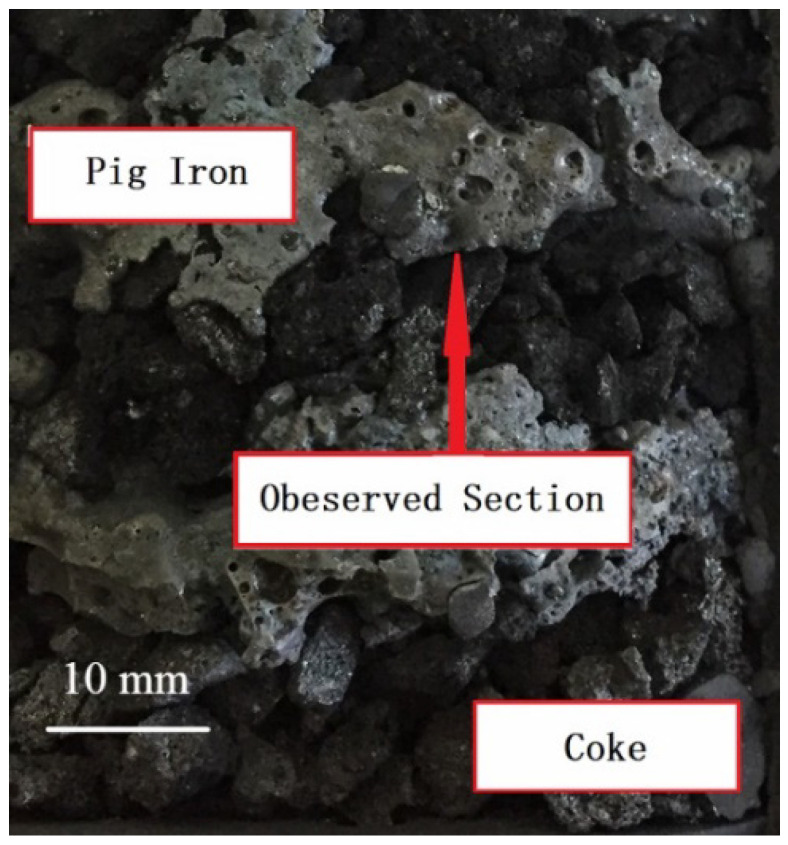
Photo of dissected graphite crucible.

**Figure 7 materials-17-01358-f007:**
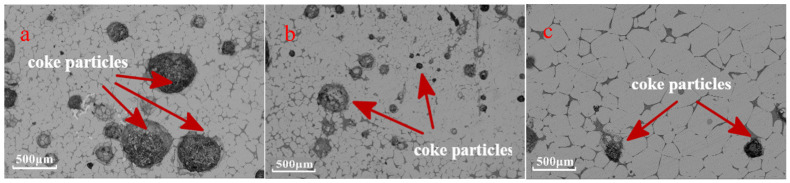
Cross section of the observed section. (**a**) Thickness 20 mm; (**b**) Thickness 30 mm; (**c**) Thickness 40 mm.

**Figure 8 materials-17-01358-f008:**
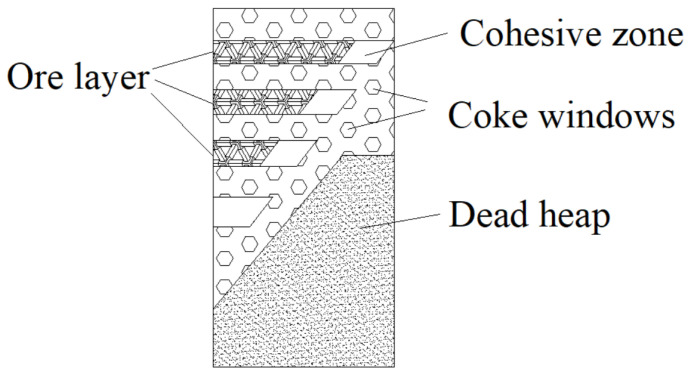
Schematic diagram of coke windows.

**Figure 9 materials-17-01358-f009:**
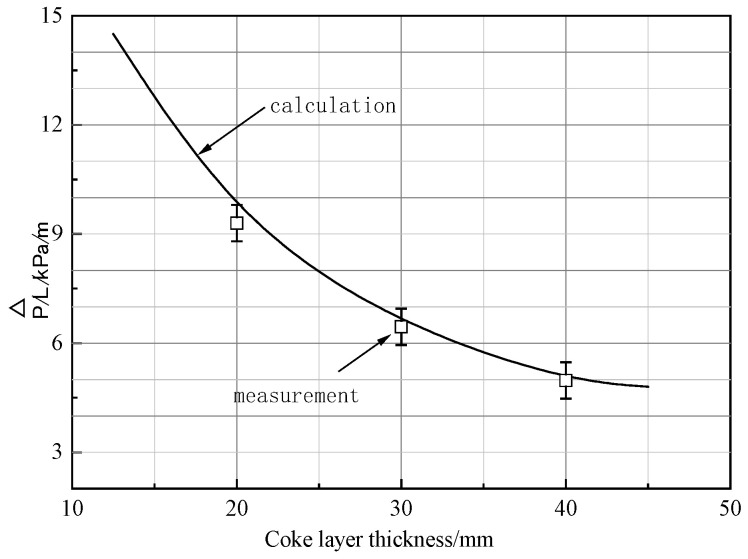
Comparison between measurement results and calculation results of Equation (7).

**Figure 10 materials-17-01358-f010:**
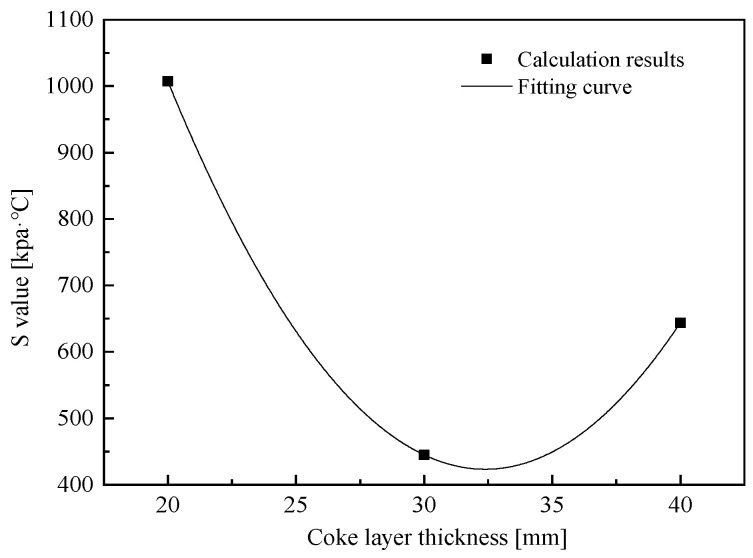
Relationship between the S value and coke layer thickness.

**Table 1 materials-17-01358-t001:** Chemical composition of raw materials, %.

Form	T[Fe]	FeO	SiO_2_	CaO	Al_2_O_3_	MgO	TiO_2_	MnO
S-1	58.52	9.33	4.85	9.04	1.69	1.47	0.11	0.22
P-1	66.53	0.32	2.02	1.49	0.59	0.16	0.04	0.13
O-1	63.39	0.48	3.14	0.04	1.25	0.06	0.04	0.13
O-2	62.47	0.32	2.96	0.04	1.26	0.05	0.04	0.27

**Table 2 materials-17-01358-t002:** Quality of coke, %.

Index	M40	CSR	Ms	A	S	C
coke	86.19	65.35	47.6	12.8	0.98	85.24

**Table 3 materials-17-01358-t003:** Experimental conditions of the materials charging.

No.	Coke Layer Thickness/mm	Packed Bed Thickness mm	Furnace Burden
①	20	104.2	65% S-1 + 20% P-1 + 15% O-1
②	30	172.5	65% S-1 + 20% P-1 + 15% O-1
③	40	241.0	65% S-1 + 20% P-1 + 15% O-1
④	20	104.5	65% S-1 + 20% P-1 + 15% O-2
⑤	40	248.4	65% S-1 + 20% P-1 + 15% O-2

**Table 4 materials-17-01358-t004:** Molten-metal dripping properties of furnace burden with different coke layer thickness.

	T_10_/°C	T_40_/°C	ΔT	T_40_/°C	P_max_	T_d_/°C	S/kPa·°C
①	1191	1278	87	1285	9322	1403	1007
②	1170	1271	101	1296	6559	1364	445
③	1169	1278	109	1289	4970	1347	644
④	1183	1282	99	1285	11,090	1360	936
⑤	1165	1280	115	1300	10,884	1365	624

## Data Availability

Data are contained within the article.

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
