# Peer review of "Study on the High-Temperature Interaction between Coke and Iron Ores with Different Layer Thicknesses"

_materials, 2024, doi:10.3390/ma17061358_

Round 1
Reviewer 1 Report
Comments and Suggestions for Authors
This work is focused on studying the effect of the thickness of the graphite layer on the performance of an experimental reactor that simulates the operation of the blast furnace. The topic is very relevant to the manufacturing process of pig iron, a fundamental raw material for steel production, which is why this article is of explicit interest to Materials readers.
I believe that before this work can be published in Materials, the following points must be addressed and corrected:
C1.- Please correct Figure 1; there is an arrow at the bottom of the reaction tower whose meaning is not explicitly expressed; apparently, it is the entry of gases into the reactor.
C2.- Clarify in the text the interest in comparing the performance of lumps o-1 and o-2 in the experimentation; what was the working hypothesis?
C3.-Please correct Figure 3. Move the vertical line of N2 2L/min towards the point (~80min, 900oC) and explain in the text the change in slope observed at approximately 1100oC in this figure.
C4.-Explain in detail how the bed shrinkage was experimentally measured using the displacement sensor.
C5.- Include a paragraph describing in detail how the parameters shown in Table 3 are phenomenologically related to the ore reduction process in the experimental device used in this work. Point out expressly in the text what each parameter indicates about what happens in the reactor.
C6.- Include a more detailed description of the results shown in Figure 4, explaining how the pressure drop is related to the events in the experiment and the performance observed in each case.
Best regards
Reviewer 2 Report
Comments and Suggestions for Authors
Comments and Suggestions for Authors
Dear Editor and Authors,
The manuscript has an important scientific and practical significance and is devoted to the issue of the interaction between coke and iron ore with different layer thicknesses.
The analysis of the manuscript shows that it needs significant improvement for possible publication in a highly indexed journal. I have pointed out the major revisions in the hope that the authors can improve the quality of the manuscript in accordance with the following comments:
1) The quality of the coke (proximate analysis, reactivity, mechanical strength) should be provided to better understand the interaction with the iron ore.
2) Why was the ratio of iron ore to coke taken as 500:80 g? Why was the diameter of ore and coke the same? The size of blast furnace coke should be more than 15 mm and is usually 25-40 mm.
3) How many repetitions for one point were performed? The repeatability of the method should be reported in the manuscript.
4) Why was the blast furnace environment modelled with CO rather than CO2 or a mixture of the two?
5) The temperature range of the pressure drop is interesting and should be additionally explained.
6) Scientific explanations are required in the manuscript along with a comparison of the results obtained with previously published results. Authors should highlight the uniqueness of the obtained results in comparison with the known ones.
7) What are the limitations of the model, taking into account the laboratory setup where the tests were conducted and the conditions in the industrial blast furnace?
8) The introduction and discussion sections of the manuscript contain few references, although the manuscript subject is well researched in scientific literature. Additionally, recent references (last 2-3 years) are completely missing. Therefore, it is recommended that the authors conduct a more detailed literature analysis for the manuscript. For example, authors may use more up-to-date references https://doi.org/10.3390/su14169947, https://doi.org/10.3390/app13179556
9) Fig. 9. The three-point curve is not representative, so it is better to use the points on the graph. Same for Fig. 10.
10) In the conclusions section, the first and third points require numerical support. The fourth paragraph needs clarification.
11) Do not end the sections with figures.
12) Use a formal style in the manuscript text. Change it’s with it is, can’t with cannot, etc.
13) The manuscript should be proofread for technical and grammatical errors that occur. Also, the terminology should be checked in the manuscript.
Comments on the Quality of English Language
The manuscript should be proofread for technical and grammatical errors that occur. The terminology should be checked in the manuscript.
Reviewer 3 Report
Comments and Suggestions for Authors
The manuscript presents a contribution to the understanding of high-temperature interactions between coke and iron ores within blast furnace operations. The detailed experimental design and analysis offer insights into optimizing blast furnace efficiency. To further enhance the manuscript, it is recommended to provide more comprehensive comparisons with existing literature, clarify methodological details, and deepen the discussion on the practical implications of the findings. Addressing these suggestions would strengthen the paper's scientific rigor and relevance. Additionally, expanding on future research directions and the broader impact of the study could offer readers a more holistic view of the research's significance within the field of metallurgical engineering and beyond. Below you can find the detailed revisions needed section-by-section.
Abstract
The abstract provides a succinct and informative overview of the research conducted on the interaction between coke and iron ores within blast furnaces, emphasizing the operational challenges posed by deteriorating ore quality. It successfully outlines the study's objectives, methods, and key findings, offering valuable insights into optimizing BF operations through the adjustment of coke layer thickness. However, the abstract could be improved by including more specific details on the research gap, methodology, quantitative results, and the broader implications of the study's findings. Enhancing the abstract in these areas would make it more informative and valuable to both researchers and industry practitioners interested in BF operational efficiency.
A1) What specific characteristics of coke and iron ore interactions at high temperatures were quantified, and how do these characteristics influence the overall efficiency of BF operations?
A2) How was the gas permeability mathematical model developed, and what were the key parameters considered in this model?
A3) What range of coke layer thicknesses was tested, and how was the "optimal" thickness determined based on the experiments?
A4) How do the metallographic microscope images contribute to understanding the coke solution loss reaction, and what were the key observations?
A5) Could the study's findings on coke layer thickness be generalized across different types of iron ore and coke materials, or are they specific to the materials tested?
1. Introduction
The Introduction section effectively sets the stage for this research by outlining the importance of coke in blast furnace operations and the specific challenges associated with managing coke layer thickness. It provides a clear rationale for the study, grounded in the operational need to optimize BF efficiency amidst deteriorating iron ore quality and the specific gap in understanding the impact of coke layer thickness. While the section successfully outlines the problem space and situates the study within a broader research context, enhancing the depth of the literature review, offering more technical specifics, and drawing clearer connections to the study's methodological approach could further strengthen it.
1.1) How does coke layer thickness specifically influence the gas flow and pressure drop within the BF, and what are the underlying mechanisms?
1.2) What previous methodologies have been employed to study the impact of coke layer thickness, and what limitations have these approaches encountered?
1.3) How do variations in coke layer thickness affect the thermal and chemical energy utilization within the BF?
1.4) What specific aspects of coke solution loss reactions are critical for understanding their impact on BF efficiency and productivity?
1.5) How does the study intend to quantify the relationship between coke layer thickness and molten metal dripping properties, and what are the expected challenges?
1.6) Some recent technology developments are missing such as: metastructures (Inverse-designed metastructures that solve equations, Science 363 (6433), 1333-1338, 2019); nanoparticles (Targeted dielectric coating of silver nanoparticles with silica to manipulate optical properties for metasurface applications, Materials Chemistry and Physics, 126250, 2022); near-zero-index materials; graphene; plasmonics and multi-functional structures. It would be beneficial for the reader if authors include such technologies in the introduction section to have a complete picture of the state-of-art.
2. Materials and Experiment
The "Materials and Experiment" section provides a foundational overview of the experimental design and materials used in the study, which is crucial for understanding the context and execution of the research. However, enhancing this section with more detailed descriptions of the experimental setup, materials selection rationale, and data collection methodologies would significantly improve its clarity, rigor, and reproducibility.
2.1) How were the specific chemical compositions of the iron ores and coke determined, and what relevance do they have to the study's objectives?
2.2) What criteria were used to design the melting-drop furnace, and how does it simulate actual BF conditions accurately?
2.3) How does the variation in coke layer thickness within the experimental setup compare to real-world variations observed in industrial BFs?
2.4) Were there any preliminary experiments conducted to optimize the experimental conditions, such as temperature and gas composition?
2.5) How was the impact of different variables controlled or accounted for during the experiments to isolate the effect of coke layer thickness?
3. High Temperature Interaction between Coke and Iron Ores
This section is crucial for advancing our understanding of the high-temperature interactions between coke and iron ores, a fundamental aspect of optimizing blast furnace efficiency and output. While the detailed technical content and specific findings are not provided here, the framework for review highlights the importance of clear data presentation, rigorous analysis, and meaningful interpretation of experimental results. Improvements in these areas, coupled with a robust discussion of the implications and future directions, can significantly enhance the scientific value and impact of the research. Ensuring that the study's findings are placed within the broader context of existing literature and industrial practice will make the research more relevant and applicable to both academia and industry.
3.1) How does the coke layer thickness specifically influence the kinetics of the high-temperature reactions between coke and iron ores?
3.2) What mechanisms were proposed or observed for the changes in gas permeability and metal dripping properties with varying coke layer thicknesses?
3.3) Were there any unexpected findings in the experimental results, and how were they addressed or explained?
3.4) How do the experimental results align with or diverge from existing theoretical models or simulations of BF operations?
3.5) What are the specific implications of these findings for optimizing blast furnace operations, particularly in terms of material selection and process parameters?
4. Discussion
The "Discussion" section is critical for articulating the significance of the study's findings within the broader field of metallurgical engineering and beyond. It should serve not only to interpret the results but also to weave them into the existing fabric of knowledge, identifying how they expand, refine, or challenge current understanding. A well-crafted discussion offers insights into the practical applications of the research, proposes directions for future investigation, and situates the study within a wider context of technological, environmental, and economic considerations.
4.1) How do the study's findings align with or challenge existing models of coke behaviour in blast furnaces?
4.2) What are the specific mechanisms through which coke layer thickness influences gas permeability and metal dripping properties, according to the study's results?
4.3) Were there any unexpected results, and how do they contribute to our understanding of high-temperature interactions in blast furnaces?
4.4) How might the limitations of the study's experimental design impact the interpretation of its findings?
4.5) What specific areas of future research does the study suggest, and how might these investigations build on the current findings?
5. Conclusion
The "Conclusion" section is crucial for wrapping up the research paper, offering a moment to reflect on the study's significance and its place within the wider body of knowledge. A well-crafted conclusion reinforces the study's contributions, articulates its practical implications, and suggests a path forward for subsequent research. While it is essential to remain concise and focused on the study's scope, providing a broader perspective on the implications can enrich the conclusion's impact.
5.1) How do the main findings of the study align with or diverge from existing theories or models in the field of blast furnace operations?
5.2) What are the specific practical recommendations for the industry based on the study's conclusions?
5.3) How might the limitations identified in the study influence the interpretation or application of the findings?
5.4) In what ways could future research build upon these findings to further enhance the efficiency and sustainability of blast furnace operations?
5.5) What broader societal or environmental impacts might arise from implementing the study's recommendations in real-world settings?
Round 2
Reviewer 1 Report
Comments and Suggestions for Authors
The authors have satisfactorily improved the document, so I believe this work can be published in Materials after addressing the following minor corrections:
C1.- Please correct scale bar in Fig. 7(b)
C2.- Please include the fitting curve in Figure 10
Author Response
C1.- Please correct scale bar in Fig. 7(b) Have been corrected. All observations must be under the same conditions, and all the scale bar in Fig 7 is 500μm. C2.- Please include the fitting curve in Figure 10 Have added the fitting curve in Figure 10.

Reviewer 2 Report
Comments and Suggestions for Authors
Dear Editor and Authors,
Unfortunately, I have to conclude that the authors have not done their best to noticeably improve the manuscript from the first report.
Information on the second point of the first report has not been added to the manuscript, although it is very important from a methodological and practical point of view. Clarifying information on the other points in the report has also not been added.
The authors did not consider it necessary to add important information to the manuscript. There is no response to the 6th point, although the uniqueness of the work and its value should be highlighted. It is not clear from the manuscript what new knowledge and results the work provides.
Conclusions do not provide numerical results and do not demonstrate the uniqueness of the work.
All indexes in tables should be deciphered. Numbering of tables should be unified.
The conclusions or title of the manuscript should highlight the laboratory conditions of the research and the results obtained.
The added explanations on page 8 refer to charcoal, which is not applicable to this paper. The previous sentences are not clear. For example, the use of the phrases 'reaction moves towards' and 'leading to coke' is unclear in meaning.
The revised manuscript is not provided in the journal template.
I believe that in its current form the manuscript is not acceptable for a highly indexed journal.
Author Response
1 Information on the second point of the first report has not been added to the manuscript, although it is very important from a methodological and practical point of view. Clarifying information on the other points in the report has also not been added.
Answer: Sorry, in the first report I have clarifying information about experimental condition but not added to the manuscript. And now I have added.
2 The authors did not consider it necessary to add important information to the manuscript. There is no response to the 6th point, although the uniqueness of the work and its value should be highlighted. It is not clear from the manuscript what new knowledge and results the work provides.
Answer: I am very sorry to forget to answer the 6th point.
Increasing the thickness of the coke layer means an increase in the ore batch, which is beneficial for strengthening smelting and has a positive effect on reducing the coke ratio and stabilizing the gas flow in the upper part of the blast furnace. However, from the results of this study, it can be seen that the beneficial effects of coke layer thickness are also limited. For different blast furnaces, the appropriate coke layer thickness should be determined based on their actual furnace conditions. The existing research and analysis focus on the analysis of blast furnace indicators after increasing the ore batch (corresponding to an increase in coke batch and coke layer thickness), and there is no mechanistic research on how the reaction between ore and coke layers affects the permeability and pressure.
Of course, further in-depth research is needed for this article.
3 Conclusions do not provide numerical results and do not demonstrate the uniqueness of the work.
Answer: have modified.
4 All indexes in tables should be deciphered. Numbering of tables should be unified.
Answer: The indexes in tables.2 have been deciphered. And numbering of tables have been unified such as table.1, table.2, ……
5 The conclusions or title of the manuscript should highlight the laboratory conditions of the research and the results obtained.
Answer: Yes, the conclusions of this experiment were all drawn under the experimental conditions of the reference national standard, and it should be explained. But it still has reference significance for actual blast furnace operation.
6 The added explanations on page 8 refer to charcoal, which is not applicable to this paper. The previous sentences are not clear. For example, the use of the phrases 'reaction moves towards' and 'leading to coke' is unclear in meaning.
Answer: I am sorry, on page 8 the charcoal is refer to coke, it is completely a mistake. And the sentences have been modified.
Reviewer 3 Report
Comments and Suggestions for Authors
Authors answered clearly to the reviewer concerns.
I suggest authors to insert ALL the answers in the manuscript properly.
New interesting applications and future works can be envisioned.
Author Response
thanks!
I have insert answers in the manuscript.
